



# Comparative characterization of bio-aerosol nebulizers in connection to atmospheric simulation chambers

Silvia G. Danelli[1,2], Marco Brunoldi[1,2], Dario Massabò[1,2,\*], Franco Parodi[2], Virginia Vernocchi[1,2], Paolo Prati[1,2]

[1]Dipartimento di Fisica - Università di Genova, via Dodecaneso 33, 16146, Genova (IT)
[2]INFN – Sezione di Genova, via Dodecaneso 33, 16146, Genova (IT)

*Correspondence to:* Dario Massabò (massabo@ge.infn.it)

**Abstract.** The interplay of bio-aerosol dispersion and impact, meteorology, air quality is gaining increasing interest in the wide spectrum of atmospheric science. Experiments conducted inside confined artificial environments, such as the

Atmospheric Simulations Chambers (ASCs), where atmospheric conditions and composition are controlled, can provide valuable information on bio-aerosol viability, dispersion, and impact. We focus here on the reproducible aerosolization and injection of viable microorganisms into an ASC, the first and crucial step of any experimental protocol to expose bio-aerosol at different atmospheric conditions. We compare the performance of three nebulizers specifically designed for bioaerosol applications: the Collison nebulizer, the Blaustein Atomizing Modules (BLAM) and the Sparging Liquid Aerosol Generator

(SLAG), all manufactured and commercialized by CH TECHNOLOGIES. The comparison refers to operating conditions and the concentration of viable bacteria at the nebulizer outlet, with the final goal to measure the reproducibility of the nebulization procedure and assess the application in experiments at ASCs. A typical bacterial test model, *Escherichia coli* (ATCC® 25922™), was selected for such characterization. Bacteria suspensions, with a concentration around $10^8$ CFU ml$^{-1}$, were first aerosolized at different air pressures and collected by a Liquid Impinger, to obtain a correlation curve between airflow and

nebulized bacteria, for each generator. Afterwards, bacteria were aerosolized inside the atmospheric simulation chamber ChAMBRe (Chamber for Aerosol Modelling and Bio-aerosol Research) to measure the reproducibility of the whole procedure. An overall reproducibility of 11% was obtained with each nebulizer through a set of baseline experiments.

## 1 Introduction

The term Primary Biological Aerosol Particles (PBAP) refers to particles of biological origins suspended in the

gaseous medium, including microorganisms and fragments of biological materials. Biological aerosol particles such as pollen, fungal spores and bacteria can cause many adverse health effects and influence earth's climate (Després et al., 2012).

Among PBAP, bacteria have a crucial role (Bowers et al., 2010). Bacteria are ubiquitous in the atmosphere, with concentrations of bacterial cells typically greater than $1\times10^4$ m$^{-3}$ over land, and, due to their small size, bacteria have a long atmospheric



residence time (about several days or more) and can be transported over long distances, up to thousands of kilometers (Burrows

et al., 2009, Federici et al., 2018).

Bacterial viability, including the capability to survive and maintain their pathogenicity, depends on several atmospheric conditions, such as relative humidity, temperature, irradiation, and chemical composition of ambient air (Marthi et al., 1990, Burrows et al., 2009). The interaction between bacteria and the other atmospheric constituents recently attracted interest as part of the broader field of atmospheric sciences (Amato et al., 2015, Brotto et al., 2015, Massabò et al., 2018). The referenced

experiments were performed by ASCs, which are small to large-scale facilities (with volumes ranging between a few to hundreds cubic meters), where atmospheric conditions can be maintained and monitored in real time for periods long enough to mimic real environments and study interactions among their constituents (Finlayson- Pitts and Pitts, 2000; Becker, 2006).

In view of long-term and systematic studies, the aerosolization of a reproducible number of viable bacteria inside an ASC is the first and not-trivial step of the experimental procedure and deserves a special attention. There are several phases potentially

critical to variability and interpretation of data (Alsved et al., 2019): preparation of the material pre-aerosolization, aerosol generation, injection and stay in the ASC, sampling, and analysis. Bio-aerosol experiments require nebulization devices that can provide high particle concentrations with minimal damage to microorganisms and their viability. The choice of the best equipment for a given application is often hampered by the lack of information on performances or drawbacks of single instruments. So far, different new aerosol generators have been designed, to improve the preservation of cultivability and

structural integrity of the aerosolized microorganisms (Zhen et al., 2014). Single-pass devices are now available, where the solution to be sprayed passes through the aerosolization nozzle just one time thus minimizing the stress to microorganisms. Market-available examples are the Blaustein Atomizing Modules (BLAM) and the Sparging Liquid Aerosol Generator (SLAG), both distributed by CH TECHNOLOGIES. The BLAM concept is an improvement of the pneumatic nebulization mechanism without liquid recirculation. The SLAG is a bubbling generator, designed for low-pressure aerosolization of

sensitive and delicate microorganisms: it implements the concept of bursting bubbles to aerosolize particles developed by Mainelis et al., 2005.

This study compares the performance of three nebulizers: the widespread Collison nebulizer (manufactured and commercialized by CH TECHNOLOGIES too), the Blaustein Atomizing Module and the Sparging Liquid Aerosol Generator. The comparison focusses on the reproducibility of operating conditions at atmospheric simulation chambers and on the

concentration of viable bacteria at the nebulizer outlet.

## 2 Materials and methods

### 2.1 Selection and preparation of bacterial strain

Gram-negative bacteria *Escherichia coli* (ATCC® 25922™) was selected as test bacterial specie. *E. coli* is a rod-shaped Enterobacter, about 1 - 2 μm long and about 0.25 μm in diameter (Jang et al., 2017). This organism is used in bio-aerosol

research as standard test bacteria (Lee et al., 2002 and 2003, Simon et al., 2011).



Prior to experiments, bacteria are cultivated on a non-selective Tryptic Soy Broth (TSB) medium until the mid-exponential phase (Optical Density, OD, at $\lambda = 600$ nm around 0.5) and then centrifuged at 4000 g for 10 minutes. Afterwards, bacteria are resuspended in sterile physiological solution (NaCl 0.9 %) to prepare a suspension of approximately $10^8$ CFU ml$^{-1}$ (Colony Forming Units), as verified by standard dilution plating. For the experiments performed inside the simulation chamber, the bacteria concentration was around $10^7$ CFU ml$^{-1}$ (see Massabò et al., 2018 for details). The average on CFU counting is used to estimate the uncertainty range of the bacterial concentration in the solutions.

## 2.2 Nebulization systems

Many natural sources of bio-aerosol arise from wet environments; bacteria and viruses are commonly found in liquid suspensions and are hence aerosolized from liquids (Alsved et al., 2019). Among the liquid bio-aerosol generators, the pneumatic nebulizers, as the Collison device, are the most frequently used. Each atomizer considered in this study works with different pressure range and aerosolization flow rate, as described below.

The Collison nebulizer has widespread applications, produces high concentrations of aerosol, but can cause damage to microorganisms due to strong impaction and shear forces. The recirculation of the cell suspension increases fragmentation of bacteria during prolonged nebulization as well (Reponen et al., 1997, Zhen et al., 2014). This device generates droplets by physical shearing and impaction onto a vessel wall. The solution to be sprayed is positioned directly in the glass jar. The compressed air is used to aspirate the liquid from the reservoir into a sonic velocity air jet, wherein the liquid is sheared into droplets. The resulting liquid jet impacts against the wall of the jar, removing the larger fraction (in size) of the droplets. The resulting smaller droplets are carried out by the airflow while the larger particles return to the liquid reservoir are then re-aerosolized. In this work, the 1 nozzle version of Collison was used. The upstream pressure can span in the 1 - 6 bar range, which corresponds to an airflow rate from 2 to 7 lpm, for the 1-jet model. The main disadvantage of this device is the recirculation of the liquid: the repetitive exposure to shear forces during atomization and impaction against the vessel wall can progressively cause damage and loss of viability to biological entities (Zhen et al., 2014). Several literature studies on the Collison performance report high particle concentrations but with a resulting cell damage (Mainelis et al., 2005; Thomas et al., 2011; Zhen et al., 2014).

The single-jet BLAM is used in one-pass mode: the liquid medium is subjected to the sonic air jet only one time. The atomizing head is composed of two main parts: nozzle body and expansion plate (Fig. 1). The atomizer features a modular design, composed of five interchangeable plates, with different cavity depth and cone diameter, to accommodate liquids with different properties (viscosity, mainly). The atomization process is generated by a vacuum effect produced in the cavity between the body of the nozzle and the expansion plate, when pressurized air passes at sonic velocity through a precisely laser cut ruby crystal (fixed size 0.010 in. diameter) located into the nozzle body (Fig. 1). This effect pushes the liquid hosted in the cavity into the air jet, which breaks up the liquid into droplets. Only the droplets smaller than a certain critical size can follow the airflow to the outlet tube located on the top of the BLAM unit: this critical size is determined by the speed of the airflow through the nebulizer. The jar should be filled with 20 ml of test solution, which serves only as a soft impaction surface for



the larger droplets and it is not used for atomization. The liquid is delivered to the nozzle body with a desired flow rate (range of liquid feed rate: 0.1 – 6 ml min-1) using a precision pump (NE-300 Just Infusion™ Syringe Pump, New Era Pump Systems, Inc.). The upstream air pressure determines the resulting airflow rate in the range 1 to 4 lpm which is kept constant by a mass flow controller. The properties of the aerosol generated by the single-jet BLAM are, nominally, a function of the jet hole size, depth of the liquid cavity, expansion cone size, and liquid viscosity. In this work, the expansion plate with a cavity depth and a cone diameter of 0.001 and 0.020 in., respectively, was used.

So far, the bubbling mechanism has been studied as a naturally occurring phenomenon and has been recognized as a significant factor in aerosolization of seawater and suspended contaminants from breaking waves (Mainelis et al., 2005). The SLAG model is a single pass bubbling generator where a suspension of particles or microorganisms is pumped at a certain flow rate to the top surface of a porous stainless-steel disk where it forms a liquid film. Then, the airflow is delivered through the porous disc creating fine bubbles in the liquid film that subsequently burst, releasing particles into the air. Particles are carried out of the device by the same air stream. We used a SLAG with a 0.75" diameter porous disk and nominal pore size of 2 µm. The recommended airflow ranges between 2 and 6 lpm and it is set by a mass flow controller. This principle of gentle bubbling aerosolization is expected to reduce stress and damage to microorganisms compared to pneumatic nebulization (Simon et al., 2006).

### 2.3 Experimental set up

In the first phase we used the experimental setup schematically shown in Fig. 2. The aerosol was sampled directly at the output of the nebulizer, through a flanged connection, by an impinging system (liquid impinger by Aquaria Srl) filled with 20 ml of sterile physiological solution and operated at a constant airflow of 12.5 lpm. The bacteria suspension (concentration about $10^8$ CFU ml$^{-1}$, see section 2.1.), was sprayed and directly collected by the liquid impinger. The number of cultivable cells inside the impinger was then determined as CFUs by standard dilution plating: 100 µl of serial dilutions of the solution was spread on an agar non-selective culture medium (trypticase soy agar, TSA), and incubated overnight at 37 °C before the CFU counting. For each nebulizer, different airflows were tested, using a mass flow controller (Bronkhorst, model F201C-FA), and the nebulization efficiency was determined in terms of culturable fraction of aerosolized bacteria (i.e., percentage ratio of the concentration of viable bacteria inside the liquid impinger and in the sprayed solution).

The further tests took place at ChAMBRe (Chamber for Aerosol Modelling and Bio-aerosol Research), a 2.2 m$^3$ stainless steel atmospheric simulation chamber specifically designed for the research on atmospheric bio-aerosol. At ChAMBRe, particles in the dimensional range of bacteria (1-2 µm) have a lifetime of several hours (Massabò et al., 2018). Atmospheric conditions and composition (i.e., type and concentration of gaseous species and PM) can be monitored and controlled. Water vapour can be directly injected into ChAMBRe thus adjusting the relative humidity inside the chamber from 0 to about 99 %. Temperature and relative humidity (RH%) inside the chamber are monitored using a HMT334 Vaisala® Humicap® transmitter. In the operative range (from 15 to 25 °C), the RH accuracy is ± 1 % RH (0 to 90 % RH) and ± 1.7 % RH (90 to 100 %RH), the temperature accuracy is ± 0.2 at 20 °C. A set of two pressure gauges is used to measure the atmospheric pressure inside and



outside the chamber. A MKS Instruments 910 DualTrans™ transducer is installed inside (measuring range from 5 x $10^{-4}$ to 2 x $10^3$ mbar; accuracy of ±0.75% of reading in the range 15 - 1000 mbar). A Vaisala BAROCAP® Barometer PTB110 is installed outside the chamber with a measuring range from 5 x $10^2$ to 1.1 x $10^3$ mbar and accuracy of ±0.3 mbar at 20 °C.

Ambient gas monitors from Environnement SA (model: O342e, AC32e, CO12e, AF22e and VOC72M) continuously measure the concentration of ozone, nitrogen oxides, carbon monoxide and dioxide, sulphur dioxide and BTEX, inside the chamber volume. Detection limits and sensitivity are 0.2 ppb and 0.1 ppb for O342e and AC32e; 0.035 ppm and 0.015 ppm for CO12e; 0.4 ppb and 0.2 ppb for AF22e and 0.05 µg $m^{-3}$ for VOC72M (reading of Benzene), with a precision of 0.025 µg $m^{-3}$ at 0.5 µg $m^{-3}$ of Benzene. Particle concentration inside the chamber is measured continuously by an Optical Particle Counter (OPC,

mod. Envirocheck 1.107, GRIMM Technologies, Inc.), which operates at 675 nm, in the 0.25 – 32 µm size range with a 6 s time resolution and a flow rate of 1.2 lpm. The OPC is periodically factory calibrated via monodisperse latex particles for size classification. A fan is installed in the bottom part of the chamber to favour the mixing of the gas and aerosol species in the reactor. Acquisition and control of devices connected to ChAMBRe are handled by a National Instruments-based system made up of a main controller (NI9057 cRIO) and several modules (C Series modules) which allow the communication with the

peripheral devices via analogical, serial and ethernet data exchange. A custom NI Labview SCADA (Supervisory Control And Data Acquisition) application allows the user to interact with the system by a user-friendly graphical interface. ChAMBRe is equipped with a sterilization system too: a 58 cm long UV lamp (UV-STYLO-F-60H, Light Progress srl) is inserted through a lateral flange. The lamp produces a 60 W UV radiation at λ = 253.7 nm which is used to sterilize the chamber volume without producing ozone before and after any experiment with bio-aerosol. Before each test with the nebulizers, the chamber was

cleaned by evacuating the internal volume down to $10^{-5}$ mbar thanks to a composite pumping system (a rotary pump model TRIVAC® D65B, Leybold Vacuum, followed by a root pump model RUVAC WAU 251, Leybold Vacuum and a Leybold Turbovac 1000). Then, the chamber was vented again to atmospheric pressure throughout a 5-stage filtering/purifying inlet (including a HEPA filter, model: PFIHE842, NW25/40 Inlet/Outlet – 25/55 SCFM, 99.97 % efficient at 0.3 µm). This filtering system ensures an excellent purification of the air entering the chamber: after venting, particles and gases concentration inside

the chamber are lower than the typical environmental values and close to the instruments sensitivity. During the experiments reported in this work, the $CO_2$ has been kept constant around 450 ppm thanks to a $CO_2$ feedback system based on a steady PID algorithm. The chamber conditions for each experiment were: temperature between 292 and 295 K, atmospheric pressure between 990 and 1020 bar, relative humidity around 70%. The fan was turned on during all the experiments, with a nominal speed of 5 rpm that results in a mixing time of about 160 s. Sets of experiments with a particular aerosol generator at fixed

setting, were performed varying the concentration of bacteria, in the physiological solution sprayed into the chamber, around $10^7$ cell $ml^{-1}$. Inside the chamber, the exposure time was up about 5 h, according to the lifetime in ChAMBRe of particles with diameter around 1 µm (Massabò et al., 2018). Finally, bacteria were collected by gravitational settling on four petri dishes, filled with trypticase soy agar medium, placed in the bottom of the chamber through an automated shelf (Massabò et al., 2018) and maintained in that position for the whole 5-hour period. Once extracted outside the chamber, the Petri dishes were

incubated overnight at 37 °C, to determine the bacteria culturable fraction by CFU visual counting. The chamber sterility was



periodically checked through a blank experiment (i.e., injecting sterile physiological solution only): no bacterial contamination has been observed in the four Petri dishes positioned on the sliding tray. Further details on the experimental protocols at ChAMBRe are reported in Massabò et al., 2018.

## 3 Results and Discussion

### 3.1 Tests with Impinger

In the first set of experiments, we measured the nebulization efficiency, in terms of culturable fraction of aerosolized bacteria for each device and at different airflows. We adopted the ratio between the CFU counted in the impinger liquid and the CFU introduced in liquid solution of the nebulizer as operative definition of efficiency. With BLAM and SLAG, the latter corresponds to the product of the concentration (i.e., CFU ml$^{-1}$) in the bacterial solution for the volume of liquid (2 ml)

introduced in the nebulizer. To have a comparable metric, in the experiments with the Collison the volume of the liquid was substituted with the injection time (5 min). Even if this choice does not meet a strict metrological criterium it makes possible a direct comparison of the three devices in well-defined operative conditions (see Table 1). The aerosolization air flow varied in the range of 1.4 - 3.5 lpm for BLAM and 2 - 5 lpm for SLAG and Collison. The bacteria suspension was supplied to the BLAM and SLAG devices at the same liquid flow rate of 0.4 ml min$^{-1}$ (see Table 1). The tests started after a suitable warming

time (about five minute), to get a stable nebulizer output. Afterwards, the aerosol was extracted for a further 5-minute time with an impinger flow of 12.5 lpm. This way, 2 ml of bacteria suspension at the flow rate of 0.4 ml min$^{-1}$ were aerosolized both by BLAM and SLAG.

Figures 3-5 show the nebulization efficiency of the BLAM, SLAG and Collison, respectively. The average on CFU counting was used to evaluate the uncertainty range of the bacterial concentration in the nebulized solution, while the uncertainty on the

air flow was determined as the 1% of the flow controller full scale. At fixed air flow, the BLAM shows the highest nebulization efficiency, followed by Collison and SLAG (e.g., at 3.5 lpm the BLAM efficiency is about 2 and 4 times higher than the Collison and SLAG ones, respectively). Our experimental procedure did not allow a direct control of the fraction of damaged bacteria during the nebulization phase, but, in the specific case of the Collison nebulizer (Fig. 5), the nebulization efficiency of the culturable fraction increases linearly with the airflow until about 3 lpm, after that the curve bends likely because the cell

damage becomes more and more relevant. However, with the described injection conditions (5 min, air flow ≤ 5 lpm) the output of viable bacteria turned out to be quite high.

At the same time, with the BLAM the flow of liquid supplied to the nebulizer can be accurately tuned. The SLAG requires a lower upstream pressure and, according to the producer claim, results in a softer injection (and then less bio-damage) of viable bacteria. Therefore, the SLAG looks best suited for experiment with fragile bacteria that can be nebulized in large numbers

even with its extremely gentle nebulization system. The BLAM efficiency seems subjected to a higher variability: such feature is likely due to the coupling between the nebulizer and the impinger set-up since the experiments with injection directly into the simulation chamber resulted much more stable (see section 3.2).



## 3.2 Tests at ChAMBRe

In the second set of experiments, we focused on the performance of the three aerosol generators when used to nebulize bacteria
directly inside an atmospheric simulation chamber. Four experiments were performed with BLAM and Collison and five with
SLAG, all between November 2019 and July 2020. Experimental conditions and results are reported in Tables 2-4. The
uncertainties quoted on both injected and collected bacteria are just those deriving from the Poisson fluctuation (i.e., the square
root of the number of colonies counted in the Petri dishes) and they do not include any other systematic or statistical term. The
values of the collected CFU are the average of the counts of the four Petri dishes exposed in each experiment; each group of
four turned out to be statistically compatible (i.e., within the interval delimited by the statistical uncertainty, the counts in the
four Petri dishes agreed). Inside the chamber, the working condition adopted for the Collison produces an initial $PM_{10}$
concentration of about 200 mg m$^{-3}$ (Table 4), like the BLAM output (Table 2). The initial $PM_{10}$ concentration, as determined
by the OPC, was taken as a rough reference for the aerosolization efficiency and quantity of aerosol generated (bacteria plus
NaCl particles).

The injected bacteria correspond to the product of the concentration (i.e., CFU ml$^{-1}$) in the bacterial solution for the volume of
liquid (2 ml) introduced in the nebulizer for BLAM and SLAG. In the experiments with the Collison, the injection time was
considered instead of the liquid volume to calculate the number of inject bacteria and thus to make possible a direct comparison
with the BLAM and SLAG performance. At ChAMBRe, considering the range of inlet air flows for the three devices, the
typical figure for the ratio between the CFU on petri dishes (diameter: 10 cm) placed inside the camber to collect the bacteria
by a gravitational settling and the injected CFU, is 10$^{-6}$ for each nebulizer. A good and stable correlation between the number
of injected and collected CFU was obtained for each nebulizer, as shown in Fig. 6-8, which refer to BLAM, SLAG and Collison
respectively. The uncertainty on the slope of the correlation curves always turned out to be < 5% and the overall standard
deviation around the average ratio (collected/injected CFU) was 11%. The experimental reproducibility appears to be adequate
to design experiments within an ASC: it roughly corresponds to the sensitivity of the whole procedure to changes in the
viability, for instance when bacteria will be exposed to different air quality conditions.

The absolute value of the aerosolization efficiency depends on the pressure at the nebulizer outlet (i.e., inside the atmospheric
chamber, Feng et al., 2020). The results presented in this work were performed in a specific pressure regime i.e., with internal
pressure about 2 mbar lower than the ambient pressure. This condition favors the bacteria confinement inside the chamber and
was explored in view of experiments with pathogenic strains. With each specific set-up (i.e., simulation chamber or other
downstream expansion volumes), the actual nebulization efficiency should be determined following the same steps above
reported. At ChAMBRe, the internal pressure can be maintained up to ± 5 mbar greater/lower than the ambient pressure. At
ChAMBRe we could verify that with a 3-5 mbar overpressure and an internal pressure ranging from 1011 to 1026 mbar, the
Collison efficiency in nebulizing physiological solution remain stable within 9%.



## 4 Conclusions

We compared the performance of three commercial nebulizer (BLAM, SLAG and Collison) in the operative conditions that could be used in experiments at atmospheric simulation chamber: one-shot injection with high output of viable biological particles. With all the instruments, the nebulization efficiency of *E. Coli* allowed to reach bacteria concentration in the order of $10^7$ CFU m$^{-3}$ in the 2.2 m$^3$ volume of the ChAMBRe ASC after a 4-5 min injection time. However, at fixed upstream air flow, the nebulization efficiency increases by a factor 2 from SLAG to Collison and from Collison to BLAM. The handling of

the devices becomes more laborious moving from Collison to SLAG and then to BLAM. Nevertheless, a set of baseline experiment at ChAMBRe (i.e., injection and suspension of *E. Coli* in a "clean" atmosphere) revealed a reproducibility of 11% regardless of the used nebulizer. Such achievement, not trivial when handling biological systems, put the basis of systematic studies on the possible correlation between bacteria viability and air quality conditions.

## Author contribution

DM, FP, and PP designed and built ChAMBRe; SGD, DM, VV and PP ran all the injections with bacteria; SGD took care of all the biological issues and measurements; MB designed and implemented the acquisition software; SGD, DM, and PP prepared the article with contributions from all of the other authors.

## Competing interests

The authors declare that they have no conflict of interest.

## Acknowledgments

The authors are indebted with the technical and administrative staff of INFN Genova for the continuous and effective support to the development of the ChAMBRe facility. This project/work has received funding from the European Union's Horizon 2020 research and innovation programme through the EUROCHAMP-2020 Infrastructure Activity under grant agreement No 730997.

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



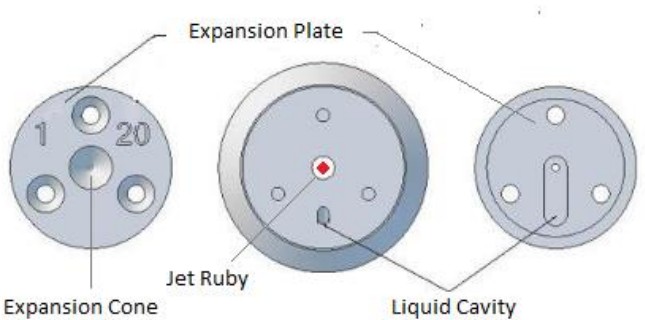

**Figure 1: Components of the BLAM Nozzle Body. Extracted and modified from BLAM user's manual.**

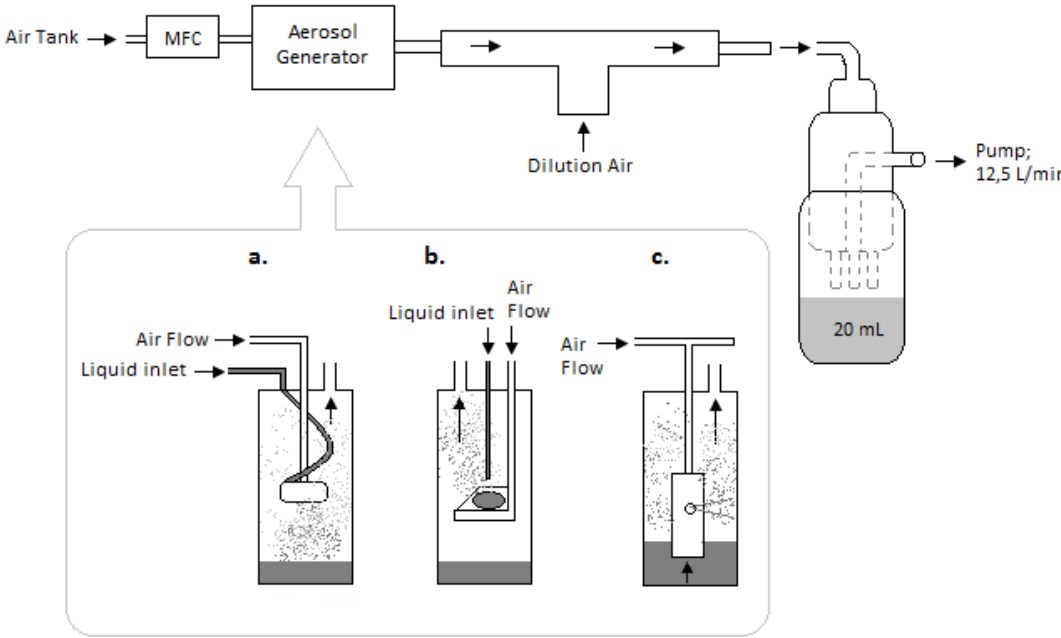

**Figure 2: Experimental setup for the tests with the impinger, a. BLAM b. SLAG c. Collison.**





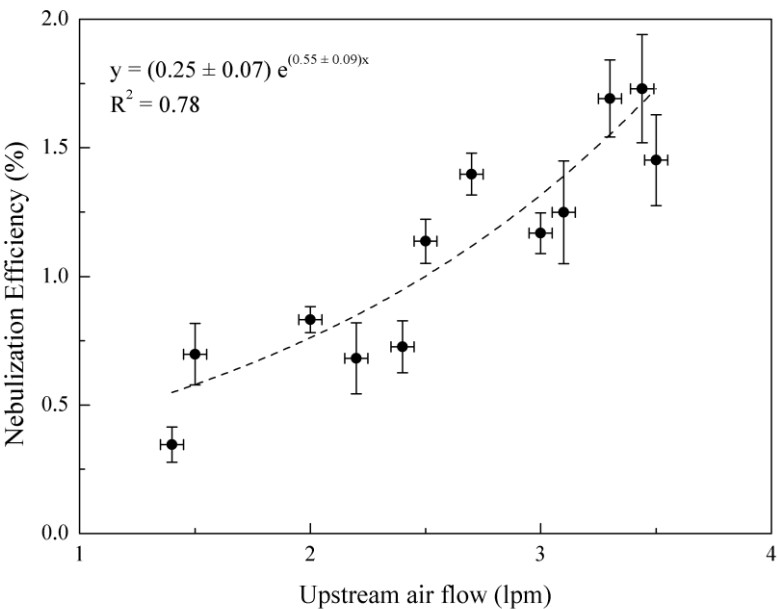

**Figure 3: BLAM nebulization efficiency vs upstream air flow. Liquid feed rate = 0.4 ml min-1.**

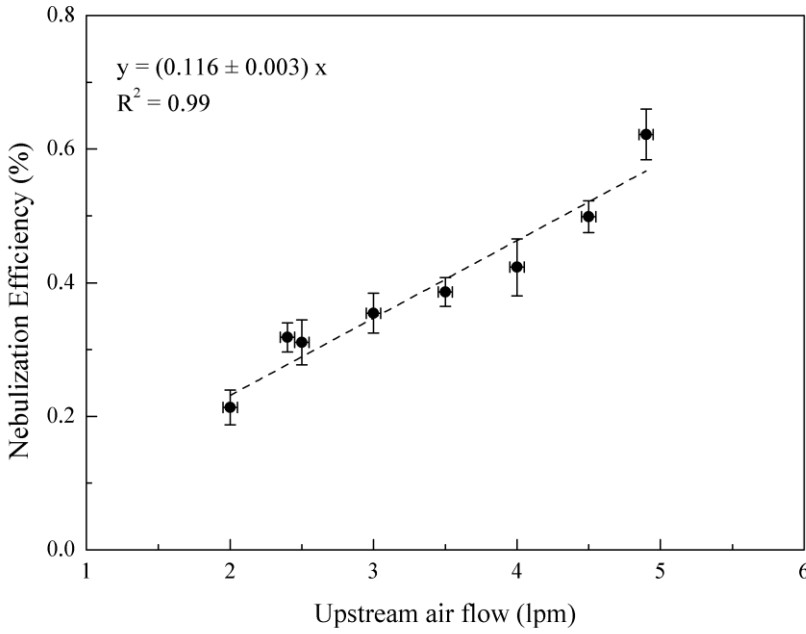

**Figure 4: SLAG nebulization efficiency vs upstream air flow. Liquid feed rate = 0.4 ml min-1.**





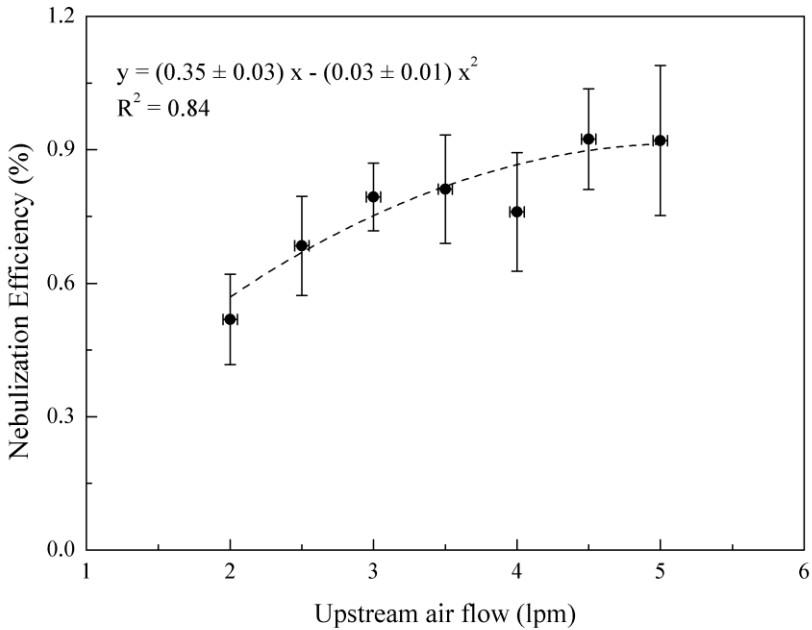


**Figure 5: Collison nebulization efficiency vs upstream air flow.**

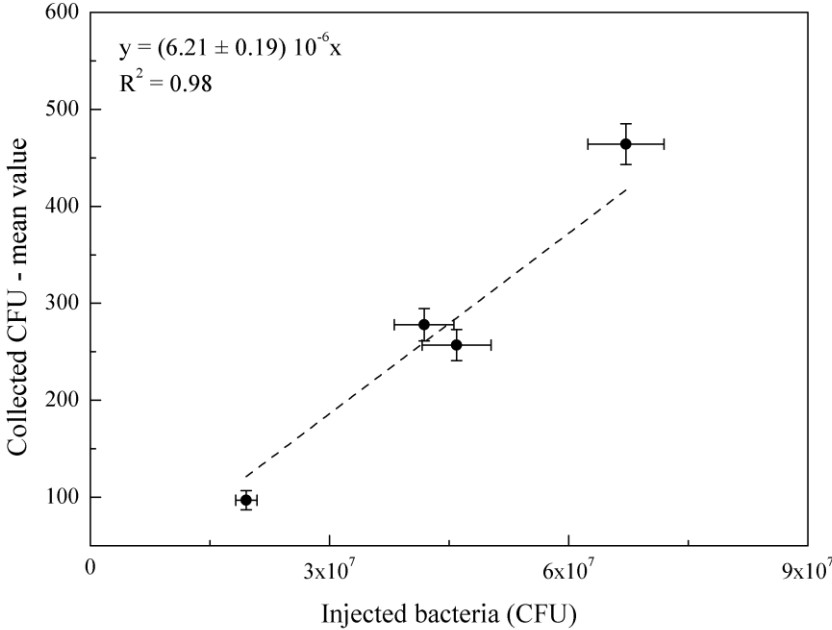

**Figure 6: Correlation curve between the number of E. coli bacteria injected in ChAMBRe by BLAM and the average count on the four Petri dishes exposed in each experiment. Injected bacteria correspond to the product of the concentration (i.e., CFU ml-1) in the bacterial suspension for the volume of liquid (2 ml) introduced into the nebulizer.**




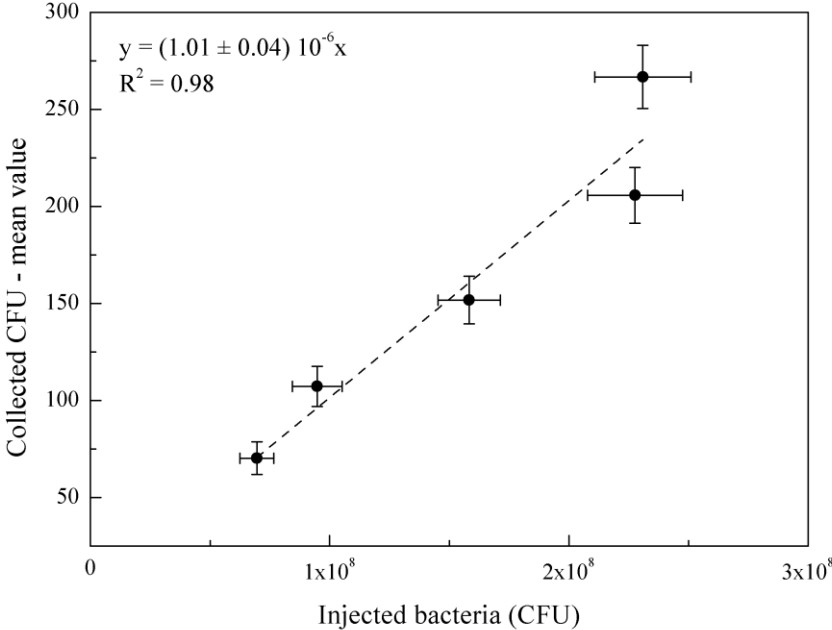

**Figure 7: Correlation curve between the number of E. coli bacteria injected in ChAMBRe by SLAG and the average count on the four Petri dishes exposed in each experiment. The injected bacteria correspond to the product of the concentration (i.e., CFU ml-1) in the bacterial suspension for the volume of liquid (2 ml) introduced into the nebulizer.**



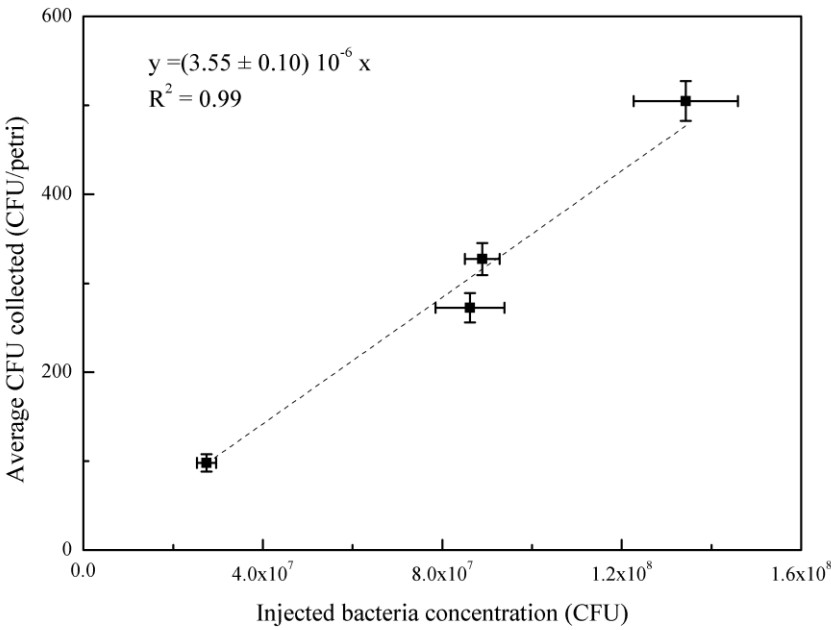


**Figure 8: Correlation curve between the number of E. coli bacteria injected in ChAMBRe by Collison and the average count on the four Petri dishes exposed in each experiment. The injected bacteria were operatively calculated as the product of the concentration (i.e., CFU ml-1) in the bacterial suspension and the injection time (4 min**





**Table 1.** Working condition of each nebulizer during the tests with liquid impinger.

| Nebulizer | Liquid Feed Rate (ml min$^{-1}$) | Volume of liquid (ml) | Injection time (min) | Air Flow range (lpm) |
|---|---|---|---|---|
| BLAM | 0.4 | 2 | 5 | 1.4 - 3.7 |
| SLAG | 0.4 | 2 | 5 | 2 - 5 |
| COLLISON | n.a. | n.a. | 5 | 2 - 5 |

**Table 2.** Bacteria concentration in the aerosolized solution, average number of colonies counted on the petri dishes and the meteorological parameters (P, T, RH) in ChAMBRe in the experiments with the BLAM nebulizer operated at: liquid feed rate = 0.4 ml min$^{-1}$, volume of injected solution = 2 ml, injection time = 5 min, air flow = 2.4 lpm.

| Date | Bacteria Concentration (CFU ml$^{-1}$) x $10^7$ | Average CFU Collected | PM$_{10}$ ($\mu$g m$^{-3}$) | External Pressure (mbar) | Internal Pressure (mbar) | Temperature (°C) | Relative Humidity (%) |
|---|---|---|---|---|---|---|---|
| 01 July 2020 | 3.36 ± 0.24 | 464 ± 21 | 190 ± 14 | 1010 | 1008 | 23.3 | 61.1 |
| 02 July 2020 | 2.09 ± 0.19 | 278 ± 17 | 170 ± 13 | 1009 | 1006 | 23.2 | 61.9 |
| 06 July 2020 | 2.30 ± 0.22 | 257 ± 16 | 190 ± 14 | 1007 | 1004 | 25.1 | 64.3 |
| 07 July 2020 | 0.98 ± 0.07 | 97 ± 10 | 190 ± 14 | 1009 | 1006 | 22.8 | 60.8 |

**Table 3.** Bacteria concentration in the aerosolized solution, average number of colonies counted on the petri dishes and the meteorological parameters (P, T, RH) in ChAMBRe in the experiments with the SLAG nebulizer operated at: liquid feed rate = 0.4 ml min$^{-1}$, volume of injected solution = 2 ml, injection time = 5 min, air flow = 3.5 lpm.

| Date | Bacteria Concentration (CFU ml$^{-1}$) x $10^8$ | Average CFU Collected | PM$_{10}$ ($\mu$g m$^{-3}$) | External Pressure (mbar) | Internal Pressure (mbar) | Temperature (°C) | Relative Humidity (%) |
|---|---|---|---|---|---|---|---|
| 18 October 2019 | 0.35 ± 0.04 | 70 ± 8 | 60 ± 8 | 1007 | 1006 | 20.7 | 57.8 |
| 19 October 2019 | 1.15 ± 0.10 | 267 ± 16 | 85 ± 9 | 1007 | 1007 | 22.8 | 59.4 |
| 20 October 2019 | 1.14 ± 0.10 | 206 ± 14 | 85 ± 9 | 1008 | 1007 | 23.2 | 63.0 |
| 28 October 2019 | 0.79 ± 0.07 | 152 ± 12 | 110 ± 10 | 998.2 | 997.7 | 22.6 | 60.0 |
| 02 December 2019 | 0.47 ± 0.05 | 107 ± 10 | 90 ± 9 | 1008 | 1008 | 21.8 | 58.1 |



**Table 4.** Bacteria concentration in the aerosolized solution, average number of colonies counted on the petri dishes and the meteorological parameters (P, T, RH) in ChAMBRe in the experiments with the Collison nebulizer operated at: liquid feed rate = 0.4 ml min⁻¹, volume of injected solution = 2 ml, injection time = 4 min, air flow = 3.0 lpm.

| Date | Bacteria Concentration (CFU ml⁻¹) x 10⁷ | Average CFU Collected | PM₁₀ (μg m⁻³) | External Pressure (mbar) | Internal Pressure (mbar) | Temperature (°C) | Relative Humidity (%) |
|---|---|---|---|---|---|---|---|
| 14 July 2020 | 2.15 ± 0.19 | 273 ± 17 | 190 ± 14 | 1010 | 1009 | 24.3 | 66.7 |
| 20 July 2020 | 2.22 ± 0.10 | 327 ± 18 | 230 ± 15 | 1013 | 1010 | 24.0 | 63.5 |
| 21 July 2020 | 0.69 ± 0.05 | 98 ± 10 | 230 ± 15 | 1016 | 1014 | 23.9 | 63.0 |
| 22 July 2020 | 0.34 ± 0.03 | 505 ± 23 | 240 ± 16 | 1014 | 1012 | 24.1 | 62.7 |