# Peer review of "Comparative characterization of performance of bio-aerosol nebulizers in connection to atmospheric simulation chambers"

_Atmospheric Measurement Techniques, 2020_

## Author Comment (AC1)

**RC1**

**General comment**

The paper focuses on an analysis of performances of nebulizers to produce airborne bacteria in laboratory experiments. The topic is interesting and it has useful application in bio-aerosol laboratory studies. The paper is well readable and suitable for the Journal. However, there are some aspects not very convincing or at least not very clear (see my specific comments). Therefore, I would suggest to consider the paper for publication after a major revision step.

We thank the Reviewer for the valuable comments, we reply point by point directly in the test (blue lines):

**Specific comments**

One aspect that is not very clear to me is that the paper is focused on the analysis of performance of nebulizers evaluated in terms of viable bacteria. I can understand the setup with the impinge that seems the one used to really evaluate the efficiencies. However, it should be explained what is the role of chamber experiments in the context of efficiencies of nebulizers. Have these experiments be used somewhat to calculate efficiencies? A discussion on this aspect is needed.

The Reviewer is right. Actually, the chamber experiments came after the set of tests with the nebulizers and aimed to assess the reproducibility of the whole procedure, having fixed a working condition (possibly the best one) for each nebulizer. Therefore, they did not add any information on the efficiency of the nebulizers.

We added the follow statements:

Line 164-165: *"The comparison here focuses on the reproducibility of the operating conditions of the nebulizers when coupled to atmospheric simulation chambers."*

 Line 202-203:*"The goal of this set of experiments was to assess of the reproducibility of the whole procedure by fixing a working condition for each nebulizer."*

Figure 3 shows efficiencies larger than one, even if by definition should be limited to one. It does not seem to be a problem of uncertainty in the counting because several points are larger than one even including the error bars. What is the explanation? An interpretation of this values should be provided in the manuscript.

We modified the Figures 3, 4 and 5 captions to make clear that the efficiency is given as a percentage values.

Figures 6, 7, and 8 show that when more viable bacteria are injected in the chamber, more viable bacteria are collected after deposition on petri dishes in similar conditions. This seems quite straightforward; I believe that a discussion explaining how the slopes are related to the efficiencies of nebulizers should be included. The differences in the slopes are due to the efficiencies of nebulizers or the injected bacteria in the x-axis have been corrected for the different efficiencies? Are they compatible with the efficiency found with the impinge setup?

As for the first comment, the text is not clear enough. Actually, the relationship between number of bacteria nebulized and number of viable bacteria collected on the petri dishes passes through the deposition losses on the walls of the chamber and through the viability reduction inside the chamber environment. The aim of these bunches of experiments was to find a quantitative and reproducible link between these quantities. The nebulization efficiency (and

its stability) is a crucial part of the game, which however remains more complex. In the three figures, the slope decreases from the BLAM, to the Collison and finally to the SLAG as the nebulization efficiency does. We decided to not correct the injected bacteria by the nebulization efficiency (but we could correct, of course) to have a value on the x-axis directly and simply determined by the bacteria concentration in the initial solution and the volume/time of injection. We consider this "operative" approach easier for the control of the whole procedure.

We have added at line 221 the following text:

*"Actually, the relationship between number of bacteria nebulized and number of viable bacteria collected on the petri dishes passes through the deposition losses on the walls of the chamber and through the viability reduction inside the chamber environment. The aim of these bunches of experiments was to find a quantitative and reproducible link between all these quantities. In Fig. 6-8, the slope decreases from the BLAM, to the Collison and finally to the SLAG as the nebulization efficiency does (see Fig. 3-5). The concentration of injected bacteria has not been corrected by the nebulization efficiency, this way values on the x-axis are directly and simply determined by the bacteria concentration in the initial solution and the volume/time of injection."*

The title should probably include the word performances, like bio-aerosol nebulizers performances" or something similar.

We agree with the Reviewer and we propose to modify the title as follow:

"*Comparative characterization of the performance of bio-aerosol nebulizers in connection to atmospheric simulation chambers*".

Line 9. Better atmospheric sciences.

Done.

Lines 65-66. This sentence is not clear. It should be added that counting errors are assumed to be equal to the square root of counting in agreement with a Poisson statistics. Or something similar.

The errors issue is discussed at the line 197. We modified the lines 66-68 as follow:

*"The average of the CFU counting is used to estimate the uncertainty range of the bacterial concentration in the solutions following the Poisson statistics (i.e., the square root of the number of colonies counted in the Petri dishes)."*

---

## Author Comment (AC2)

**RC2**

Very interesting paper because the generation methods are always needed for the experimental works in aerosol science.

We thank the Reviewer for the valuable comments, we reply point by point directly in the test (blue lines):

I have only few remarks to help the readers.

Line 63 NaCl 0,9 % is it by volume or by weight?

The concentration of the physiological solution (i.e. normal saline) is commonly expressed by weight per unit volume. We changed the text writing: NaCl 0.9 % w/v.

CFU ml$^{-1}$ should give how many CFU in air in the best conditions?

This actually refers to the concentration of CFU in the physiological solution (to be sprayed in the chamber).

Page 3 line 80 Collison nebulizer can be supplied in non-recirculation mode with a syringe pump.

The Reviewer is right however, we had such limitation with our nebulizer. We added the Reviewer comment in the revised text as follow:

Line 85: *"It is worthy to note that, differently from the specific model used in this work, some Collison units can be operated in not-recirculation mode by a syringe pump."*

Line 90 inch is imperial unit. Why not cm?

We modified the text as follow: *"..(fixed size 0.025 cm diameter).."*

Page 5 line 140  too many different units are used  cfm; standard CFM; lpm;  °C;  K. I would suggest to keep lpm (1cfm = 28.4 lpm) and °C rather than Kelvin.

Thanks for the note: we made uniform the units throughout the text.

Page 6 line 175  blam slag produce how many particles /cc ?

With the impinger we assume to collect all the nebulized particles in the 20 ml of physiological solution inside the impinger. The concentration of viable particles (i.e. CFU) in the solution therefore depends on the liquid volume and we did not consider this number too informative. We couldn't measure the number of total bacteria (i.e. viable and not-viable) and/or other particles in the nebulizers output flow.

We could roughly estimate the conc. of injected viable bacteria in the chamber but we did not have the possibility to directly measure/control losses on the walls and on the inlets. Through the text, we refer the collected CFUs (both in impingers and in the petri dishes inside the chamber) to the CFU concentration in the injected solution x the injection volume/time.

Line 209 cambre should be chamber.

Done.

Page 4 line 113 remove the dot after 2.1

Done.

Page 4 line 126. The temperature accuracy unit is not given. Is it 0.2% or 0.2 °C?

The temperature accuracy unit is ±0.2 °C at 20 °C. We have modified the text accordingly.

Page 5 and others. The pressure is given in mbars. This unit is not legal unit. The pressure must be given in Pascal. The authors can add between parenthesis mbar if they want.

We have modified as suggested the pressure units.

Page 5 line 149. 'The pressure in the ChAMBRe arise from 10-5 mb to atmospheric pressure with air (I guess)' . A precision should be given about this air? Is it atmospheric air (called lab air) or air from a cylinder? If lab air is used then the authors should precise the RH. Indeed it seems that they are not using any drying system.

As the Reviewer points out we use "lab air", i.e. we have a drying/filtering system, which reduces the R.H. to about 15%. We added this information in the text.

Page 5 line 153. The pressure given is little bit incorrect 990 and 1020 bars are too high as pressure. I guess that the unit is mbars (again).

Yes, it was a material mistake that we corrected (and we moved to IS units).

Page 7 line 203. What it means PM10? The size distribution is monitored with an optical particle counter. That will be very nice to give more details on the measured distributions since the OPC gives them. Are they reproducible? What is the sigma g of the distributions? What is the density value used to calculate this relatively high mass concentration (200 mg/m$^3$) from the number concentrations given by the OPC? 200mg/m3 seems monumental form me.

We are sorry for a second material mistake: actually, we got 200 microg/m3 and not millig/m3, We corrected the value in the revised text. The OPC signal is dominated by the NaCl particles of the physiological solution: The size distribution turned out to be quite stable even if quite disperse. In one single experiment with the BLAM (not quoted in the text), we observed by a SMPS that the large majority of the particles are smaller than 100 nm.

The sampling experiments in the ChAMBRe are conducted by gravitational settling. The gravitational settling of a particle of 1 μm is 3.5 10-5 m/s in still air. I think that your method penalizes the generator. I would prefer a single stage bio impactor if I had to carry these experiments.

In our chambers, the lifetime of 1-micron particles is around 5 hours (Massabò et al., 2018) and we designed our experiments accordingly. In the future, we'll also test the use of bioaerosol impactors.

The short conclusion of the paper is not giving the results of each generator clearly. It will be better to give the concentration 5CFU/m3 for each generator to help the reader. It will be good to recall the concentrations at the outputs of each generator always to help the reader.

See our answer above to the Reviewer question at page 6, line 175. In the present configuration we cannot measure the bacteria concentration inside the chamber and therefore we anchored to the reference of CFU in the injected solution times the injection volume. We

could quote the CFU concentration at the nebulizers outlet but we could be misleading since it could be just proportional to the final concentration inside the chamber volume. The aim of our experiments was to assess the reproducibility of the whole procedure while ensuring a good statistics for the counting on the petri dishes placed at the bottom of the chamber (and finally to demonstrate the sensitivity to possible changes of bacteria viability with polluted atmospheres). We consider such achievement as a good starting point for further future improvements.

I would suggest to add one paper at least on bio aerosols and atmosphere (for example Joung 2017 : Bioaerosol generation by raindrops on soil:  Nature communications 8 : 14668)

We added the reference at line 71.

---

## Author Comment (AC3)

**RC3**

It is a very interesting paper that compares the performance of three nebulizers for the work of bio-aerosols in atmospheric simulation chambers (ASC). The authors investigated the efficiencies of the nebulizers in association with the airflow and the subsequent viable fraction of the bacterial cells after nebulization into an ASC. The paper points out nicely the advantages and disadvantages regarding the performances of each nebulizer and allows the reader to follow easily by pointing out a clear conclusion in the end.

We thank the Reviewer for the valuable comments, we reply point by point directly in the test (blue lines).

Just a few remarks from my side:

Make sure to use the same font, size, ect. to fullfill the formart requirements

Thanks for the note: we uniformed the format throughout the text.

It's difficult to understand what the 11% (line 22) refer to exactly. Make it more clear in both the abstract and results section.

The Reviewer is right. We added the following sentences:

Lines 21-22: "*(i.e. standard deviation of the results obtained with the three nebulizers)*"

Lines 213-215: "*This value corresponds to the standard deviation of the results of the entire bunch of experiments around the mean value of the collected to injected CFU ratio (taking into account the results of all the three nebulizers).*"

Line 39: "non-trivial"

Done.

Line 65: "of the cfu..."

Done.

I think it is very nice how you described the different nebulizers in 2.2, however it would make more sense to move this section to the Introduction section

We see the point however, since the different performance/behavior of the three nebulizers is a central aspect of our technical paper we'd prefer to maintain this description in the "material and method" section.

Line 142: "sterilization system, too"

Done.

In section 3.1 it would be nice to read a bit more about how it is possible to compare one nebulizer in ml, while the other one is in minutes

The Review is right and we are aware that we used a quite peculiar "metric". We added the following sentences at line 171- 174, to clarify the reason of this choice:

*"Since the Collison nebulizer is working in a recirculation mode, (i) it is not possible to quantify the absolute value of the liquid volume passing through the nozzle, (ii) an unknown fraction of the liquid to be aerosolized pass through the nozzle more than once. To have a comparable metric, in the experiments with the Collison the volume of the liquid was substituted with the injection time (5 min, the same used with other two nebulizers)".*

Avoid statements that are vage such as in line 186: "output of viable bacteria turned out to be quite high" and rather write those results with clear statements (eg. by including numbers)

The Review is right and we had modified the sentences at line 186 as follow:

*"However, with the described injection conditions (5 min, air flow ≤ 5 lpm) the output of viable bacteria turned out to be comparable with the results obtained with the other two nebulizers."*

Use the same units throughout the whole paper

Thanks for the note: we uniformed the units throughout the text.